# Toxicity of Silver Nanoparticles Supported by Surface-Modified Zirconium Dioxide with Dihydroquercetin

**DOI:** 10.3390/nano12183195

**Published:** 2022-09-14

**Authors:** Dušan Sredojević, Vesna Lazić, Andrea Pirković, Jovana Periša, Natalija Murafa, Biljana Spremo-Potparević, Lada Živković, Dijana Topalović, Aleksandra Zarubica, Milica Jovanović Krivokuća, Jovan M. Nedeljković

**Affiliations:** 1Centre of Excellence for Photoconversion, Vinča Institute of Nuclear Sciences—National Institute of the Republic of Serbia, University of Belgrade, 11000 Belgrade, Serbia; 2Department of Science, Texas A&M University at Qatar, Doha P.O. Box 23874, Qatar; 3Department for Biology of Reproduction, INEP Institute for Application of Nuclear Energy, University of Belgrade, 11000 Belgrade, Serbia; 4Institute of Inorganic Chemistry of the Czech Academy of Sciences, 250 68 Husinec-Řež, Czech Republic; 5Department of Pathobiology, Faculty of Pharmacy, University of Belgrade, 11000 Belgrade, Serbia; 6Department of Chemistry, Faculty of Science and Mathematics, University of Niš, Višegradska 33, 18000 Niš, Serbia

**Keywords:** zirconium dioxide nanoparticles, dihydroquercetin, silver nanoparticles, toxicity, antimicrobial ability

## Abstract

The antibacterial performance and cytotoxic examination of in situ prepared silver nanoparticles (Ag NPs), on inorganic-organic hybrid nanopowder consisting of zirconium dioxide nanoparticles (ZrO_2_ NPs) and dihydroquercetin (DHQ), was performed against Gram (−) bacteria *Escherichia coli* and Gram (+) bacteria *Staphylococcus aureus*, as well as against human cervical cancer cells HeLa and healthy MRC-5 human cells. The surface modification of ZrO_2_ NPs, synthesized by the sol-gel method, with DHQ leads to the interfacial charge transfer (ICT) complex formation indicated by the appearance of absorption in the visible spectral range. The prepared samples were thoroughly characterized (TEM, XRD, reflection spectroscopy), and, in addition, the spectroscopic observations are supported by the density functional theory (DFT) calculations using a cluster model. The concentration- and time-dependent antibacterial tests indicated a complete reduction of bacterial species, *E. coli* and *S. aureus*, for all investigated concentrations of silver (0.10, 0.25, and 0.50 mg/mL) after 24 h of contact. On the other side, the functionalized ZrO_2_ NPs with DHQ, before and after deposition of Ag NPs, do not display a significant decrease in the viability of HeLa MRC-5 cells in any of the used concentrations compared to the control.

## 1. Introduction

Dihydroquercetin (DHQ), also known as Taxifolin, is a flavonoid extracted from Siberian or Dahurian larch with significant pharmacological actions, including anticancer, antioxidative, anti-inflammatory, antigenotoxic, anti-Alzheimer, antiangiogenic, diuretic, and prevention of diseases such as cardiovascular, and diabetes [1,2,3]. Also, it is reported that DHQ enhances the antitumor effect of IFN-γ and prevents the occurrence of hepatocellular carcinoma [4]. The extraordinary properties of DHQ can positively affect nanomaterials used in dentistry and medicine and provide an additional value to nanocomposites. Zirconium dioxide (ZrO_2_) is one of the most used materials in dentistry and orthopedics.

Zirconium dioxide is a wide bandgap semiconductor (E_g_~5 eV) that attracted considerable attention from the research community due to various potential applications, including fuel cell technology, catalysis, optical filters, and nano-electronic devices [5]. However, the most promising use of ZrO_2_ is related to bio-applications since zirconia has a low affinity to bacterial biofilm formation, reduces inflammatory infiltration, and zirconia substrates show a better adhesion and proliferation of osteoblast cells than TiO_2_ substrates [6,7]. Also, the in vitro cytotoxicity of zirconia is negligible [7,8,9,10,11]. One of the most common applications of ZrO_2_ is medical, focused on bone tissue engineering and orthopedic implants, which are susceptible to biofilm formation [12]. Bacterial infections are the main reason for the failure of implants, so the application of modified materials repulsive to bacterial growth is necessary to enhance the lifetime of the medical implants. To tackle this problem, researchers came up with the idea of using biocompatible antimicrobial agents against pathological fungi and bacteria. Knowing that ZrO_2_ nanoparticles are biocompatible, have a large specific surface area, strong corrosion resistance, and affinity for oxygen-containing groups [5], the further improvement of their properties, leading to new potential applications, is possible to achieve by surface modifications with bio-active small organic molecules such as DHQ and antimicrobial agent silver nanoparticles.

The synthetic methodology for the preparation of inorganic-organic hybrids, based on bio-related materials such as TiO_2_ [13,14,15,16,17,18,19,20], hydroxyapatite [21,22], and CeO_2_ [23,24], is well-established by our group. The coordination of ligands, including many biologically active molecules such as caffeic acid, dopamine, salicylic acid, 5-aminosalicylic acid, ascorbic acid, and vitamin B6, to oxide surfaces, is facilitated by the condensation reaction between hydroxyl groups from inorganic and organic counterparts of hybrids. Consequently, the interfacial charge transfer (ICT) complex between oxide and colorless organic molecules is formed, indicated by enhanced optical properties, i.e., the red absorption shift. So far, DHQ has never been used to facilitate the ICT complex formation with any inorganic component.

An additional advantage of surface modification is the possibility of introducing free functional groups with the capability to participate in the formation of the more complex nanostructure. For example, free amino groups from ligand molecules coordinated to oxide particles’ surfaces or functionalized polymers can reduce silver ions and link synthesized Ag nanoparticles (NPs) to hybrid materials [21,25,26,27]. Of course, the purpose of preparing supported Ag NPs is clear since, from ancient times, silver was recognized as an antimicrobial agent, although its toxic mechanism on microbial species and effect on human cells is not fully understood. On the other hand, using deprotonated hydroxyl groups, only synthesis of colloidal free-standing Ag NPs was achieved [28,29].

This study aims to evaluate the antibacterial and cytotoxic activity of nanocomposite materials: (a) surface-modified ZrO_2_ with DHQ (ZrO_2_/DHQ) with improved optical properties due to the formation of the ICT complex, and (b) ZrO_2_/DHQ decorated with Ag NPs (ZrO_2_/DHQ/Ag), prepared by the reduction of Ag^+^ ions using deprotonated free hydroxyl groups from DHQ. The antibacterial performance of prepared hybrids is estimated using Gram (−) bacteria *Escherichia coli* and Gram (+) bacteria *Staphylococcus aureus.* The cytotoxic action of prepared samples against human cells was evaluated using human cervical carcinoma (HeLa) cells and normal human fetal lung fibroblast (MRC-5) cells. These cell lines are universal reference cellular models in toxicity screenings of nanomaterials [30,31]. Before bio-tests, a thorough characterization of the nanocomposite materials is carried out, including transmission electron microscopy (TEM), X-ray diffraction analysis (XRD), and UV-Vis spectroscopy. The density functional theory (DFT) calculations are applied to get a deeper insight into the optical properties and electronic structure of the ICT complex between ZrO_2_ and DHQ.

## 2. Experimental

### 2.1. Synthesis and Characterization of ZrO_2_ Nanoparticles and ZrO_2_-Based Nanocomposites

All chemicals were of the highest available purity (J.T. Baker) and used without further purification. Milli-Q deionized water with a resistivity of 18.2 MΩ cm^−1^ was used as a solvent.

The sol-gel method was applied to synthesize the ZrO_2_ nanoparticles by the hydrolysis of organic precursors. In brief, zirconium (IV) isopropoxide, dissolved in 2-propanol, was combined with the water−2-propanol solution (volume ratio 40:1); before mixing, solutions were stirred vigorously for 1 h. The precipitation of zirconia at room temperature was facilitated by adjusting the pH to 9.5 by adding the ammonia solution (25% NH_4_OH). The concentration of zirconia sol was 0.5 M. Precipitate, separated by centrifugation, was washed with deionized water several times, and at the end, with ethanol. Finally, the powder was dried and homogenized with 0.5 M H_2_SO_4_ and, after sulfurization, again dried at 120 °C for 3 h.

The ZrO_2_ nanoparticles, functionalized with dihydroquercetin (pure DHQ, commercial product of Sibpribor Ooo, Irkutsk, Russia) and decorated with Ag nanoparticles, were prepared in two steps. In the first step, ZrO_2_ nanoparticles were surface-modified with DHQ by dispersing 200 mg of ZrO_2_ nanoparticles in 50 mL of water containing 315 mg of DHQ. The dispersion was stirred at 40 °C for 24 h, and the appearance of yellow-brown color indicated the successful formation of an inorganic-organic hybrid. Then, the surface-modified powder was separated by centrifugation, washed five times with deionized water to remove excess ligand, and dried at 40 °C in the vacuum oven. In the second step, ZrO_2_/DHQ nanocomposite, decorated with Ag nanoparticles, was prepared by reducing silver ions with deprotonated hydroxyl groups from DHQ. The pH was adjusted to around 9 using the NaOH solution (0.1 M) to ensure the deprotonation of hydroxyl groups from DHQ. In a typical synthesis, the mixture of 100 mg of ZrO_2_/DHQ and 7.9 mg of AgNO_3_ in 50 mL of de-aerated water with Ar was stirred overnight at 60 °C under reflux. The synthesized nanocomposite ZrO_2_/DHQ/Ag was separated by centrifugation, washed several times with deionized water, and dried at 40 °C in a vacuum oven.

The content of silver in ZrO_2_/DHQ/Ag nanocomposite was determined using the Thermo iCAP ICP-MS instrument. Before measurements, samples were digested in a Milestone Start D microwave using concentrated HNO_3_ and H_2_O_2_.

High-resolution transmission electron microscopy (HRTEM) was applied to evaluate the morphology and structural details of the prepared materials. The HRTEM operating at an accelerating voltage of 300 kV and using cathode LaB_6_ as an electron source provides a theoretical resolution of 1.7 Å. The microscope is equipped with CCD GATAN MULTISCAN (model 794) and EDS spectrometer (INCA x-stream module OXFORD). For microscopy examination, samples were dispersed in ethanol, and a drop of the very dilute suspension was placed on a holey-carbon-coated copper grid and dried at ambient temperature.

The optical properties of synthesized powders were investigated by diffuse reflectance spectroscopy (Shimadzu UV-Visible UV-2600 spectrophotometer equipped with an integrated sphere ISR-2600 Plus).

The crystal structure of the powder was analyzed by X-ray diffraction (XRD) using a Rigaku SmartLab diffractometer (Cu-Kα_1,2_ radiation, λ = 0.1540 nm) at ambient temperature. The diffraction patterns were recorded in 2θ range from 10 to 90° with a 0.02° step and 1°/min counting time.

### 2.2. Computational Details

We performed all density functional theory (DFT) and time-dependent density functional theory (TD-DFT) calculations using the Gaussian 09 suite of programs [32]. To examine the electronic and optical properties of the ICT complex formed upon the adsorption of DHQ on the ZrO_2_ surface, we employed the [Zr_11_O_20_(OH)_4_] cluster as a model system for the calculations. This cluster was generated based on the crystal structure of the monoclinic lattice of ZrO_2_ (*P2_1_/c*), according to the (111) plane [33]. The ground-state geometry of the [Zr_11_O_20_(OH)_2_]/DHQ cluster was optimized at a gas phase with the help of CAM-B3LYP (Coulomb-attenuating method) functional [34], which considers long-range correction. Since zirconium is a transition element with atomic number 40 ([Kr]4d^2^5s^2^), we used a basis set with effective core potential (ECP) to shorten the computational time and implicitly include the relativistic effects of the inner core electrons. We chose the Stuttgart-Dresden small-core ECP basis set [35], in which 28 inner electrons are described by an effective core potential while the outer 12 electrons occupy the valence space. The Pople’s valence double-ζ polarized 6-31G(d,p) basis set was used for light elements [36]. The Zr- and O-atoms of the cluster were frozen during the optimization to preserve the crystal (monoclinic) structure of ZrO_2_, while the DHQ’s atoms were allowed to relax. The same functional has been used within the time-dependent DFT by considering the first 30 excitations to calculate the vertical excitation energies (E_vert_), the nature of transitions, and the oscillator strengths (ƒ) [37]. The optimized geometry of the cluster was obtained by the GaussView software, while the electronic excitation spectra and DOS/PDOS diagrams were derived from the GaussSum program [38,39].

### 2.3. Antibacterial Evaluation

The antibacterial ability of ZrO_2_/DHQ/Ag nanocomposite was evaluated against pathogenic bacteria species, Gram-negative bacteria *Escherichia coli* (ATCC 25922) and Gram-positive bacteria *Staphylococcus aureus* (ATCC 25923). The fresh microbial inoculum was prepared by growing microorganisms in 3 mL of tryptone soy broth (TSB) overnight at 37 °C. For the antibacterial tests, the ZrO_2_/DHQ/Ag powder (20, 50, and 100 mg) was transferred to 10 mL of saline solution (8.5% NaCl) containing diluted microbial inoculum (~10^5^ CFU/mL). The 1 mL aliquot of incubated solution at 37 °C, at sampling times of 2 and 24 h of incubation, was transferred to the sterile Petri dish and overlaid with tryptone soy agar. The inoculated plates were incubated at 37 °C, and after 24 h of bacteria growth, cells were finally counted. The surface-modified ZrO_2_ powder with DHQ did not display any significant antibacterial activity.

### 2.4. In Vitro Biological Activity

#### 2.4.1. Preparation of Stock and Working Solutions with NPs

The freshly prepared nanopowders were used for the biological tests. The stock solutions were prepared by dispersing the 10 mg/mL of nanopowders (ZrO_2_/DHQ and ZrO_2_/DHQ/Ag) in a complete cell culture medium containing RPMI 1640 Medium, with L-Glutamine (Capricorn; Ebsdorfergrund, Germany) supplemented with 10% fetal bovine serum (Sigma-Aldrich, St. Louis, MO, USA) and 1% penicillin-streptomycin (Sigma-Aldrich, St. Louis, MO, USA)). Then, suspensions were left for 24 h at 37 °C in a humidified incubator with 5% CO_2_. Upon 24 h of incubation, supernatants, separated from powders by centrifugation at 300× *g* for 5 min and subsequent filtration through the 0.22 µm filter, were used for the preparations of the treatments. Both stock solutions, diluted with a fresh cell culture medium, were used to prepare working stock solutions for the individual experiments at different nanopowders’ concentrations (0.5, 1, 2, 5, and 10 mg/mL).

#### 2.4.2. Evaluation of Cytotoxicity of ZrO_2_/DHQ and ZrO_2_/DHQ/Ag Nanoparticles

Cytotoxicity of ZrO_2_/DHQ and ZrO_2_/DHQ/Ag nanopowders at five different concentrations (0.5, 1, 2, 5, and 10 mg/mL) were evaluated using MTT, as described previously [40]. Also, the cytotoxic effects of two prepared hybrids were examined using Human Cervical Adenocarcinoma (HeLa, ATCC^®^ CCL-2™, Rockville, MD, USA) and human fetal lung fibroblast (MRC-5, ATCC^®^ CCL-171™, Rockville, MD, USA) cell lines. After thawing, HeLa and MRC-5 cells were grown as a monolayer in RPMI 1640 Medium with L-Glutamine (Capricorn; Ebsdorfergrund, Germany) supplemented with 10% fetal bovine serum (Sigma-Aldrich, St. Louis, MO, USA) and 1% penicillin-streptomycin (Sigma-Aldrich, St. Louis, MO, USA) at 37 °C, and 5% CO_2_. The medium was changed every 48 h. After reaching 70% confluence, the cells were harvested from flasks by trypsinization, using 0.25% trypsin-EDTA solution (Capricorn Scientific, Ebsdorfergrund, Germany) and seeded in 96-well plates (1.5 × 10^4^ cells/well) in 100 µL of the complete medium. Before treatment, the cells were allowed to adhere to wells for 24 h at 37 °C in a humidified incubator with 5% CO_2_. After 24 h of incubation, the medium was removed and replaced with a fresh medium containing hybrid samples up to a total culture volume of 100 µL/well, and again incubated at 37 °C for 24 h. Finally, cells were washed with PBS after removing the medium with nanopowders. For the MTT assay, 100 µL of fresh medium containing 10 µL of MTT reagent (thiazolyl blue tetrazolium bromide, 1 mg/mL) was added to each well. The cells were then incubated for 2 h in the dark at 37 °C to allow the formation of formazan crystals. Further, formazan crystals were solubilized by adding 100 µL sodium dodecyl sulfate (10% SDS, 0.1 N HCl) to each well, and plates were kept at 37 °C for 24 h before the reading. Finally, absorbance was read at 570 nm by a microplate reader (BioTek ELx800, Santa Clara, CA, USA).

### 2.5. Statistical Analysis

The data were analyzed using the ANOVA one-way analysis of variance with the Tukey posthoc test (α = 0.05). The values are expressed as mean ± standard error, and the differences were considered statistically significant at *p* < 0.05. Statistical analysis was performed using GraphPad Prism 6.0 (GraphPad Software, Inc., San Diego, CA, USA).

## 3. Results and Discussion

### 3.1. Characterization of ZrO_2_, ZrO_2_/DHQ, and ZrO_2_/DHQ/Ag

The wide-angle XRD patterns of unmodified ZrO_2_ powder and ZrO_2_/DHQ/Ag nanocomposites are presented in Figure 1. The diffractogram of ZrO_2_ displays the presence of two broad peaks at around 31° and 51° that belong to (111) and (220) planes of the cubic crystalline structure of ZrO_2_ (COD Card No. 1525707). The additional diffraction peaks in the XRD pattern of ZrO_2_/DHQ/Ag nanocomposite at 38.1, 44.3, 64.5, and 77.4 belong to (111), (200), (220), and (311) crystal planes of face-centered-cubic silver, respectively (COD Card No. 9013050).

TEM analysis of the unmodified ZrO_2_ nanopowder (Figure 2a) revealed snowflake-like loose agglomerates consisting of small-sized 2–5 nm nanoparticles. High-resolution TEM images (Figure 2b) indicated that the small nanoparticles within each agglomerate share a close crystallographic alignment. Analysis of the selected area electron diffraction (SAED) pattern (Figure 2c) revealed the presence of the diffraction rings consistent with the (111), (200), (220), and (311) planes of the cubic ZrO_2_ crystal structure. TEM images of functionalized ZrO_2_ nanopowder with DHQ are shown in Supporting Information (Appendix A). The presence of the small organic molecules on the surface of ZrO_2_ did not influence their morphology. The TEM images from ZrO_2_/DHQ/Ag nanocomposite with low- and high-magnification, and SAED patterns, are shown in (Figure 2d–f, respectively). The results indicate the random distribution of Ag nanoparticles within agglomerated snowflake ZrO_2_ particles. The size of the silver nanoparticles ranges from 5 to 20 nm. The SEAD pattern indicated the presence of the diffraction rings of silver consistent with cubic structure; diffraction rings are labeled in Figure 2f.

The content of silver in ZrO_2_/DHQ/Ag nanocomposite, determined by the ICP-OES technique, is 5 wt.-%. Although the ICP-OES technique is not selective toward the chemical state of silver, taking into consideration the thorough washing procedure of the synthesized sample, we expect that the measured value corresponds to the content of the metallic silver in the ZrO_2_/DHQ/Ag composite.

The Kubelka-Munk transformations of diffuse reflection spectra of unmodified ZrO_2_ powder, surface-modified ZrO_2_ powder with DHQ, and ZrO_2_/DHQ powder decorated with Ag NPs, are shown in Figure 3 (curves from left to right, respectively). Also, photographs of ZrO_2_, ZrO_2_/DHQ, and ZrO_2_/DHQ/Ag powders are included as an inset to Figure 3. As expected, unmodified ZrO_2_ powder shows the typical absorption in the UV spectral range, below 250 nm, due to the electronic transition from the valence to the conduction band [41]. The functionalization of ZrO_2_ powder with DHQ induced an immediate appearance of yellow color. The appearance of absorption in the visible spectral range in surface-modified ZrO_2_ powder with DHQ (absorption onset at 530 nm, i.e., 2.35 eV) is the consequence of the formation of interfacial charge transfer (ICT) complex between the surface of ZrO_2_ and a small colorless organic molecule with a specific molecular structure. While the optical properties and potential applications of the ICT complexes between TiO_2_ and salicylate- [14,17,20], catecholate- [13,14,15,16,18,19,42], and phenol-type [43,44] ligands are described in detail in the literature, to the best of our knowledge, there are no reports concerning the ZrO_2_-based ICT complexes, except for our recent study about the toxicity of surface-modified ZrO_2_ nanoparticles with caffeic acid [6].

The deposition of Ag NPs onto ZrO_2_/DHQ powder induced color change from yellow to yellow-brown (inset to Figure 3). However, the surface plasmon resonance band is not present in the reflection spectrum, most likely due to the overlap of optical absorption of Ag NPs with charge transfer transitions of the ICT complex between ZrO_2_ and DHQ.

### 3.2. DFT Calculation

We performed DFT and TD-DFT calculations to supplement experimental findings and examine the effects of surface modification of *m*-ZrO_2_ nanoparticles with DHQ molecules. The DFT methods can provide an atomistic picture of the surface-modified NPs and enable a detailed description of their electronic structures. In this study, we explored the electronic structure of the ZrO_2_/DHQ ICT complex using a cluster model, calculated frontier molecular orbitals (FMO) that govern their optical properties and compared them with experimental data. According to our previous findings, we presumed bidentate bridging coordination between DHQ and zirconium atoms placed at the (111) surface [6] as a consequence of a condensation reaction between surface hydroxyl groups of ZrO_2_ NPs and DHQ’s hydroxyl groups.

The optimized geometry of the [Zr_11_O_20_(OH)_2_]/DHQ cluster is presented in Figure 4, indicating, as already mentioned, bridging bidentate coordination between the DHQ molecule and the surface Zr-atoms. Also, we tried to optimize mononuclear chelate coordination, but this geometry gradually converged into the bridging bidentate. It is noticeable that the DHQ molecule is not perfectly perpendicular to the (111)-surface but is bent at an angle of 81° (Figure 4). The lengths of two Zr-O_DHQ_ bonds are 2.053 and 2.005 Å. The calculated bandgaps of [Zr_11_O_20_(OH)_4_] and [Zr_11_O_20_(OH)_2_]/DHQ clusters are 3.77 and 2.98 eV, respectively. The estimated bandgap value of the [Zr_11_O_20_(OH)_4_] cluster (3.77 eV) is close to the calculated bandgap value of *m*-ZrO_2_ by LDA and GGA approximation (3.58 eV [45]), but both values are far from the experimental ones in this study or the literature [41]. The further improvement of calculation techniques to achieve a better estimation of bandgap requires the inclusion of on-site Coulomb interaction (LDA + *U*) through 4d orbitals on Zr atoms (*U*^d^) and of 2p orbitals on O atom (*U*^p^) [46]. However, in this study, we intended to demonstrate that the coordination of DHQ molecules to the ZrO_2_ surface leads to the red-shifted absorption induced by the formation of the ICT complex rather than reproducing the exact value of bandgaps.

The density of states diagrams (DOS/PDOS) and the FMO for [Zr_11_O_20_(OH)_4_] and [Zr_11_O_20_(OH)_2_]/DHQ clusters representing unmodified ZrO_2_ powder and surface-modified ZrO_2_ powder with DHQ are presented in Figure 5, respectively. The valence band maximum (VBM) and conduction band minimum (CBM) approximated with HOMO/LUMO orbitals of the *m*-ZrO_2_ cluster are located at −6.42 and −2.65 eV, respectively (Figure 5A). The discrepancy between calculated and experimentally measured bandgap values is already discussed. The donor level of the [Zr_11_O_20_(OH)_2_]/DHQ clusters is instilled within the bandgap region and represents the HOMO orbital of DHQ. This level is positioned at −5.84 eV, comprising π-orbitals of the aromatic rings and p-orbitals of two bridging O-atoms (Figure 5B). The calculated HOMO/CBM gap of the [Zr_11_O_20_(OH)_2_]/DHQ clusters is 2.98 eV and is comparable to the experimentally determined absorption onset of the surface-modified ZrO_2_ powder with DHQ (2.98 versus 2.35 eV). However, the calculated HOMO/CBM gap is more related to the electrochemical bandgap mainly because, in photoexcitation, there is the creation of an electron-hole pair, which interacts by electrostatic Coulombic forces making the optical bandgap in general smaller. The CBM is approximated by the LUMO orbital of the clusters, mainly composed of 4d-orbitals of Zr-atoms, whereas VBM is mainly composed of 2p-orbitals of the oxygen atoms. In addition, the LUMO orbital of DHQ is positioned far above the CBM of *m*-ZrO_2_. Thus, the observed broad absorption band of ZrO_2_/DHQ powder in the visible spectral region (Figure 3) can be associated with the ICT transitions from the DHQ’s donor levels to the CB of ZrO_2_. The HOMO-1 and lower orbitals of DHQ occupy the VB space, and they are intertwined with the orbitals of the cluster.

The theoretical spectra of the [Zr_11_O_20_(OH)_2_]/DHQ cluster and its constituents (free DHQ molecule and [Zr_11_O_20_(OH)_4_] cluster) are shown in Figure 6. Electronic excitations of the ZrO_2_ cluster and DHQ molecule are in the UV spectral region and do not overlap with ICT transitions of [Zr_11_O_20_(OH)_2_]/DHQ complex (absorption onset ~500 nm). The data concerning relevant energy levels of DHQ, [Zr_11_O_20_(OH)_4_], and [Zr_11_O_20_(OH)_2_]/DHQ complex (VBM, HOMO, CBM, and LUMO) with the first three excitations are summarized in Table 1. By examining the wave-functions and orbital contributions of the relevant electronic excitations of the [Zr_11_O_20_(OH)_2_]/DHQ cluster, it is evident that the most pronounced transition arises from DHQ’s HOMO to CBM of the cluster (414 nm, ƒ = 0.59). The ligand HOMO is located inside the bandgap region of the [Zr_11_O_20_(OH)_4_] cluster, whereas ligand LUMO is far in the CB region (Table 1). The absorption onsets of the calculated electronic excitation spectrum of [Zr_11_O_20_(OH)_2_]/DHQ and the experimentally observed in the reflection spectrum of the corresponding surface-modified ZrO_2_/DHQ powder are at almost the same position (~500 nm) (compare Figure 3 and Figure 6).

### 3.3. Antibacterial Activity and Toxicity of ZrO_2_/DHQ/Ag

To evaluate differences in the antibacterial activity against two bacterial species (Gram-negative bacteria *Escherichia coli* and Gram-positive bacteria *Staphylococcus aureus*), exposed to ZrO_2_/DHQ/Ag nanocomposite, concentration-dependent microbial cell survivors counting was performed. The obtained results concerning the antimicrobial activity of ZrO_2_/DHQ/Ag nanocomposite against bacterial species *E. coli* and *S. aureus* are shown in Figure 7. The ZrO_2_/DHQ/Ag concentration was varied (2, 5, and 10 mg/mL), while the contact time between nanocomposite and bacterial species was 2 and 24 h. The content of Ag in ZrO_2_/DHQ/Ag nanocomposite is about 5 wt.-%, so the actual concentration of Ag was much smaller, 0.10, 0.25, and 0.50 mg/mL for the composite’s concentrations of 2, 5, and 10 mg/mL, respectively. In the control sample (ZrO_2_/DHQ), after 24 h of contact, an insignificant increase of bacterial colonies is noticeable. According to the literature data, DHQ itself displays antimicrobial activity against *E. coli* at a concentration of 5 mg/mL [47]. However, in our experiments, for the highest concentration of composite, the weight percentage of DHQ is one order of magnitude lower. After 24 h of contact, there are no live cells of either bacterial species, *E. coli* and *S. aureus*, for all investigated concentrations of nanocomposites. When the contact time between bacteria and nanocomposite was shorter (2 h), destruction of *E. coli* cells was achieved only with the highest ZrO_2_/DHQ/Ag concentration (10 mg/mL). For lower concentrations of nanocomposite (2.0 and 5.0 mg/mL), the survival number of *E. coli* cells was 20 and 10, respectively. However, after 2 h of contact, in the entire concentrations range of ZrO_2_/DHQ/Ag, the composite displays satisfactory antibacterial activity against *S. aureus* since the number of live cells was around 10^2^. The antibacterial activity of Ag NPs supported by functionalized ZrO_2_ with DHQ against *E. coli* and *S. aureus* is at the same level as the antimicrobial activity of Ag NPs attached to different supports [21,25,26,27,48,49,50,51].

### 3.4. Toxicity of ZrO_2_, ZrO_2_/DHQ, and ZrO_2_/DHQ/Ag

The results presented in Figure 8 show the effects of the ZrO_2_/DHQ and ZrO_2_/DHQ/Ag on the cell viability of human cervical carcinoma cell line (HeLa) and human fetal lung fibroblast cells (MRC-5) used as normal non-cancerous cells. The presented results indicate that exposure of HeLa cells to 24 h treatment with the ZrO_2_/DHQ and ZrO_2_/DHQ/Ag at the 5 different concentrations (0.5, 1, 2, 5, and 10 mg/mL) did not induce significant cytotoxic effects in the samples exposed to ZrO_2_/DHQ compared to the control, while the slight percentage reduction of live cells (14%) was seen only in the highest concentration. The HeLa cells incubated with ZrO_2_/DHQ/Ag did not show a significant decrease in the cell viability in any of the used concentrations versus the control.

Further, a similar effect was seen in MRC-5 cells after the 24 h exposure to the corresponding concentrations of 0.5, 1, 2, 5, and 10 mg/mL ZrO_2_/DHQ and ZrO_2_/DHQ/Ag, where neither type of nanoparticles significantly influenced the percentage of live cells compared to the control (Figure 8B). Only a slight non-significant decrease in cell viability (9%) was induced by the exposure to the ZrO_2_/DHQ at the concentration of 10 mg/mL.

The effect of nanoparticles on tumor cells has been the focus of investigation in the last decade, with an emphasis on discovering new effective antitumor drugs. Since differently prepared NPs have different characteristics that could affect their biological activity, elucidation of the mechanism of NPs’ interaction with mammalian cells requires defining the impact of their size, form, surface charge, and stabilizing agent on cell viability and function. Different types of mammalian cells have various biological responses to the same nanomaterials, and malignant cells have different responses to NPs from normal cells due to altered morphology and phenotypes [52]. Hence, to obtain the selective effects in cancer cells and produce minimum toxicity to healthy surrounding cells warrants therapeutic potential and further investigation of nano-drugs. Recent studies suggested that phytochemicals as natural antioxidants in food could alleviate nanoparticle (NP) toxicity and act protectively in one type of cells, and at the same time be cytotoxic in other types of cells, which may indicate cell-type dependent responses to combined exposure of NPs and phytochemicals [53]. Our research aimed to demonstrate the effects of newly synthesized ZrO_2_/DHQ and ZrO_2_/DHQ/Ag NPs on cell viability of human cancer cell line HeLa and normal human lung fibroblast cells MRC-5. HeLa cells represent human cervical carcinoma cells and are frequently applied as a reference cancer cell line in toxicity screenings of new nanomaterials [30]. MRC-5 lung fibroblasts are widely used for research as a healthy non-malignant cell line to test the cytocompatibility of new nanomaterials [54,55]. Thus, we evaluated the effects of the nanocomposite materials in the human healthy cells MRC-5 and human cancer HeLa cells. The obtained effects reflected the same response to the ZrO_2_/DHQ and ZrO_2_/DHQ/Ag NPs in both types of cells, and there were no cytotoxic effects observed in any of the used concentrations, for both types of NPs, except for the highest concentration of ZrO_2_/DHQ.

Earlier toxicity studies on various human cell lines have found that Ag NPs might be cytotoxic in HeLa cells at minimal concentrations (toxicity thresholds) in the range of 0.5–2.0 μg of Ag per mL [56]. Significant literature data describing the improved cytotoxic and anti-proliferative effects of the different biosynthesized Ag NPs in HeLa cells suggest the anticancer potential of green-Ag NPs [57,58,59,60]. However, in our current work, the hybrid ZrO_2_/DHQ/Ag NPs did not show improved effects when compared to the ZrO_2_/DHQ, and it seems that the addition of the Ag reduced the cytotoxicity of the NPs, probably by reducing the oxidative potential. In our previous work, we demonstrated that exposure of the HeLa cells to the bare ZrO_2_ nanoparticles does not affect cell viability, while ZrO_2_ nanoparticles functionalized with caffeic acid exhibited a small toxic effect [6]. Other authors showed that the biosynthesized ZrO_2_ NPs with plant extracts have improved bioactivity and show high antioxidant, antimicrobial activity, and cytotoxicity in human cancer cell lines [61,62]. In our current work, hybrid ZrO_2_ NPs functionalized with DHQ did not exhibit significant cytotoxic effects, indicating that the functionalization of ZrO_2_ with this polyphenol did not alter their cytotoxic potential. Recent studies suggested that phytochemicals as natural antioxidants in food could alleviate nanoparticle toxicity and act protectively in one type of cells, and at the same time be cytotoxic in other types of cells, which may indicate cell-type dependent responses to combined exposure of NPs and phytochemicals [53]. In terms of ZrO_2_, this material is considered biocompatible, and it was shown that surfaces covered with ZrO_2_ increased/accelerated activation of factors involved in the formation of new fibroblast cells and formation of connective tissue [63]. In agreement with this, we found that hybrid ZrO_2_ NPs functionalized with DHQ were not cytotoxic in human fibroblasts MRC-5.

Previous research on Ag NPs showed conflicting results in MRC-5 cells. The work of Dastgir et al. showed that treatment of MRC-5 cells with silver nanoparticles at concentrations of 25, 50, and 100 μg/mL after 24, 48, and 72 h induced a significant reduction in the viability of cells and a dose-dependent inhibitory effect on normal MRC-5 cells [64], which is consistent also with the work of determined IC50 at a concentration of 31.5 µg/mL for chemically synthesized Ag NPs in MRC-5 cells [65]. The cytotoxicity effects were also observed for the biosynthesized Ag NPs by red macroalgae *Laurencia caspica*; the IC50 value of Ag NPs after 48 h treatment for T47D and MRC-5 cell lines are 29.37 mg/mL and 42.13 mg/mL, respectively [66]. However, these concentrations exceed those used in our study. Belteky et al. observed an important phenomenon that the toxicity of Ag NPs decreases with increasing particle aggregation and that there is a massive impact of the corona effect on the biological activity of nanoparticles [67]. Because of that, they studied the influence of Ag NPs aggregation on cytotoxicity in MRC-5 cells and indicated that the colloidal stability of bare Ag NPs is affected in complex systems. So, the large-scale Ag NPs aggregation occurs near physiological electrolyte concentration, resulting in the loss of biological activity of Ag NPs. Also, Belteky et al. showed that Ag NPs synthesized with green tea extract could attenuate aggregation due to electrostatic interactions even in complex cell culture media, while Ag NPs prepared by this method retained a certain degree of toxicity. In addition, cytotoxicity was time-dependent, and cytotoxic effects are observable 3 h after incubation with NPs, while after 24 h of incubation with the same NPs, cytotoxic effects are non-existent. This effect can be explained by the initial formation of a large and less compact biomolecule corona, adsorbed on the nanoparticle surface, referred to as the “soft corona”, peaking at around 3 h of incubation, and followed by the decrease of values due to shifting to a smaller, more compact state, where the absorbed biomolecules have a higher affinity toward the nanoparticle surface, producing less cytotoxicity. Since we treated MRC-5 cells with NPs for 24 h, most likely, the reduced cytotoxic activities of ZrO_2_/DHQ and ZrO_2_/DHQ/Ag NPs are due to the formation of the compact biomolecular corona from the culture medium.

## 4. Conclusions

Both hybrid materials, ZreO_2_/DHQ and ZrO_2_/DHQ/Ag, are prepared for biomedical applications. The surface modification of ZrO_2_ with DHQ leads to the formation of covalent bonds between inorganic and organic constituents of the hybrid and, in addition, provides the possibility to in situ synthesize Ag NPs, taking advantage of remaining free hydroxyl groups. Undoubtedly, the enhanced optical property induced by the formation of the ICT complex is beneficial for photo-induced catalytic reactions and worth studying. However, our primary intent was the evaluation of the antibacterial and cytotoxic activity of prepared hybrids. Based on the obtained data, the conclusions of this study are the following:(a)Efficient antibacterial action of the ZrO_2_/DHQ/Ag hybrid against *E. coli* and *S. aureus*. Complete reduction of bacterial strains after 24 h of contact, even for the lowest concentration of hybrid (2 mg/mL, i.e., 0.1 mg/mL of Ag).(b)Neither hybrid, ZrO_2_/DHQ nor ZrO_2_/DHQ/Ag, in investigated concentration range (0.5–10 mg/mL), is cytotoxic in human cervical carcinoma (HeLa) cells and normal human fetal lung fibroblast (MRC-5) cells.

## Figures and Tables

**Figure 1 nanomaterials-12-03195-f001:**
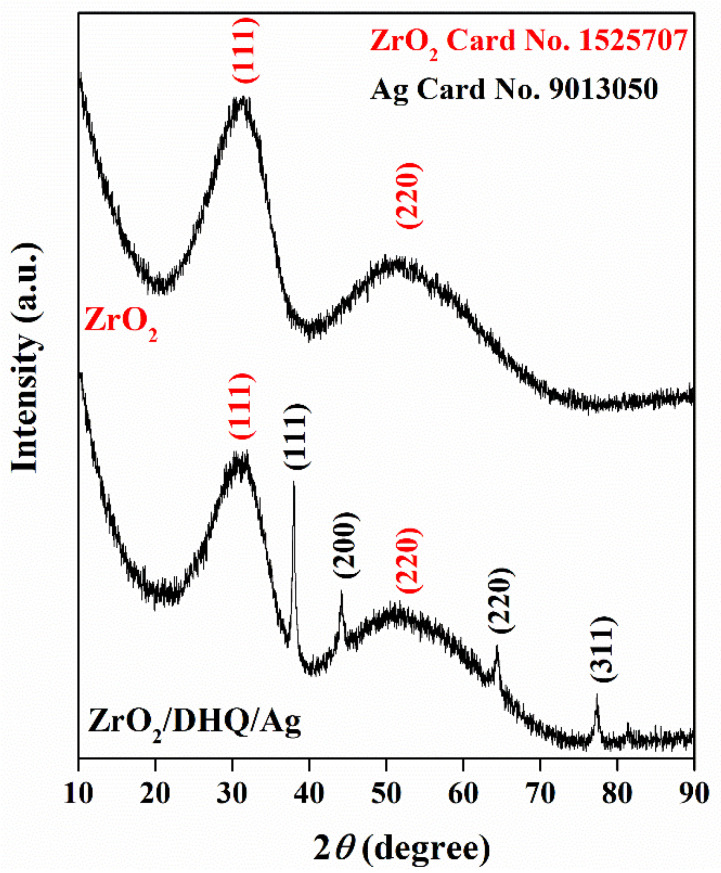
The XRD patterns of unmodified ZrO_2_ powder and ZrO_2_/DHQ/Ag nanocomposite.

**Figure 2 nanomaterials-12-03195-f002:**
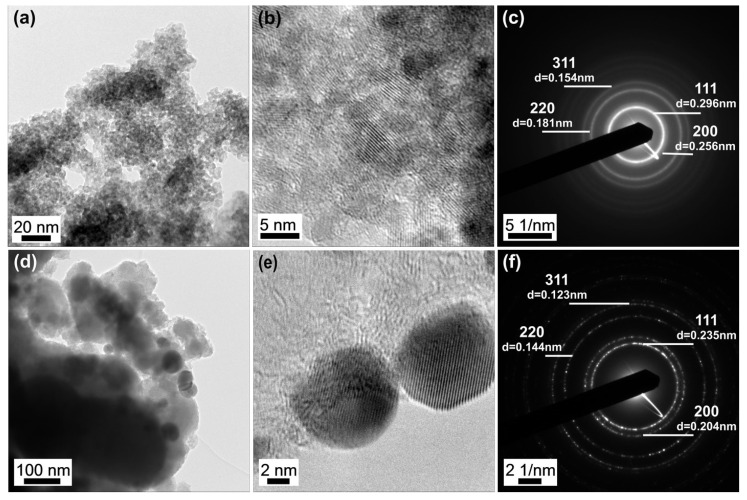
TEM images at different magnifications of unmodified ZrO_2_ (**a**,**b**) and ZrO_2_/DHQ/Ag powders (**d**,**e**) and corresponding SEAD patterns (**c**,**f**, respectively).

**Figure 3 nanomaterials-12-03195-f003:**
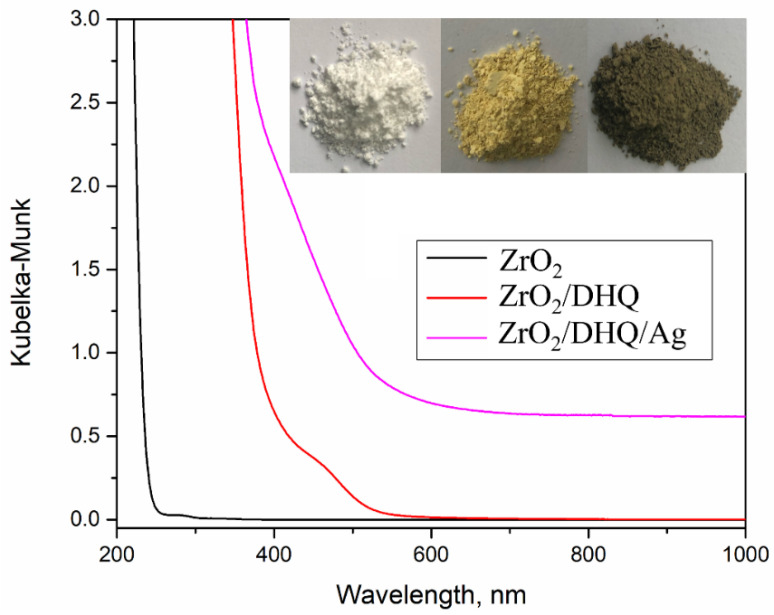
Kubelka-Munk transformations of UV-Vis-NIR diffuse reflection spectra of ZrO_2_, ZrO_2_/DHQ, and ZrO_2_/DHQ/Ag powders. Inset: photo images of ZrO_2_, ZrO_2_/DHQ, and ZrO_2_/DHQ/Ag from left to right.

**Figure 4 nanomaterials-12-03195-f004:**
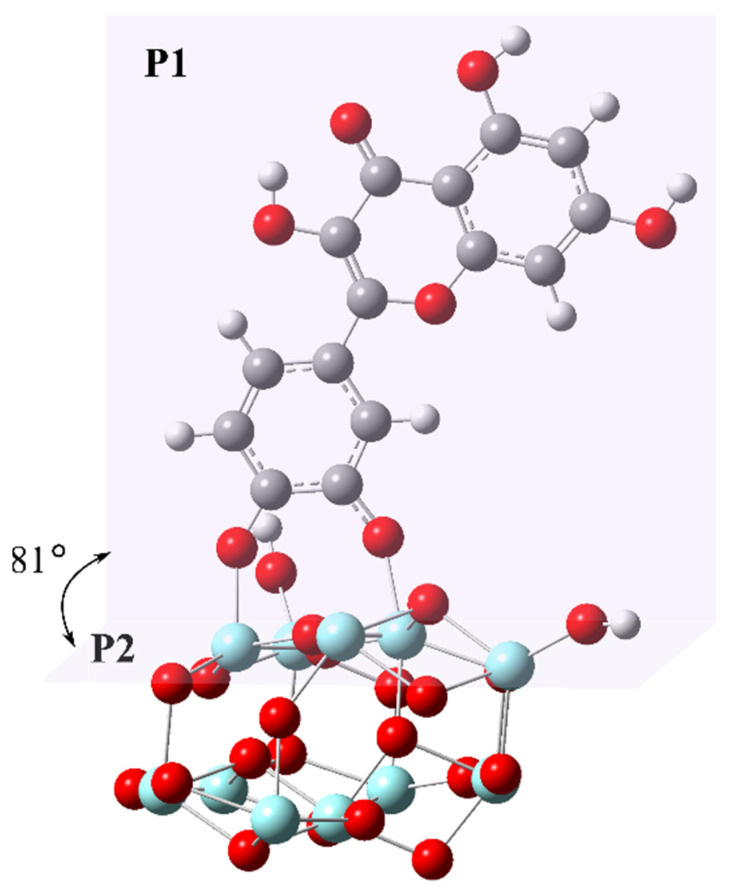
The optimized structure of [Zr_11_O_20_(OH)_2_]/DHQ cluster at the CAM-B3LYP/6-31G(d,p)/SDD(28) level of theory. The Zr, O, C, and H atoms are shown by pale blue, red, gray, and white circles. The mean planes of the coordinated DHQ molecule and (111) plane of the cluster are marked as P1 and P2, respectively.

**Figure 5 nanomaterials-12-03195-f005:**
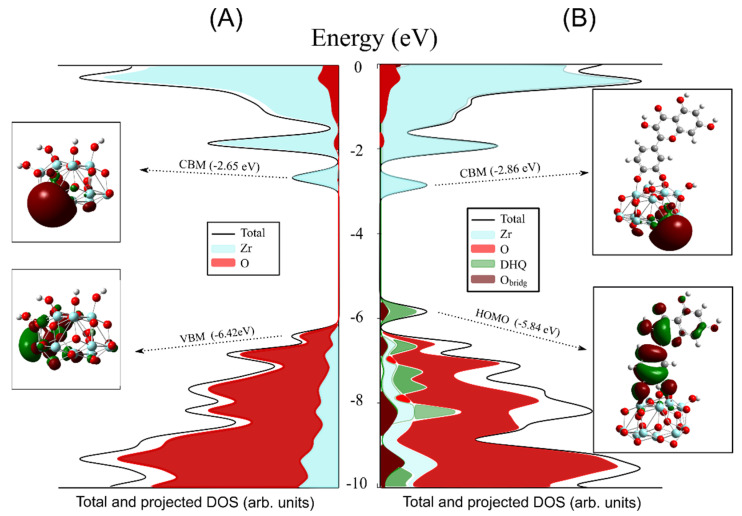
Total and projected density of states (DOS/PDOS), with FMO distributions of (**A**) [Zr_11_O_20_(OH)_4_] and (**B**) [Zr_11_O_20_(OH)_2_]/DHQ clusters, as calculated at CAM-B3LYP/6-31G(d,p)/SDD(28) level. Gray: carbon, white: hydrogen, red: oxygen, and pale blue: zirconium atoms.

**Figure 6 nanomaterials-12-03195-f006:**
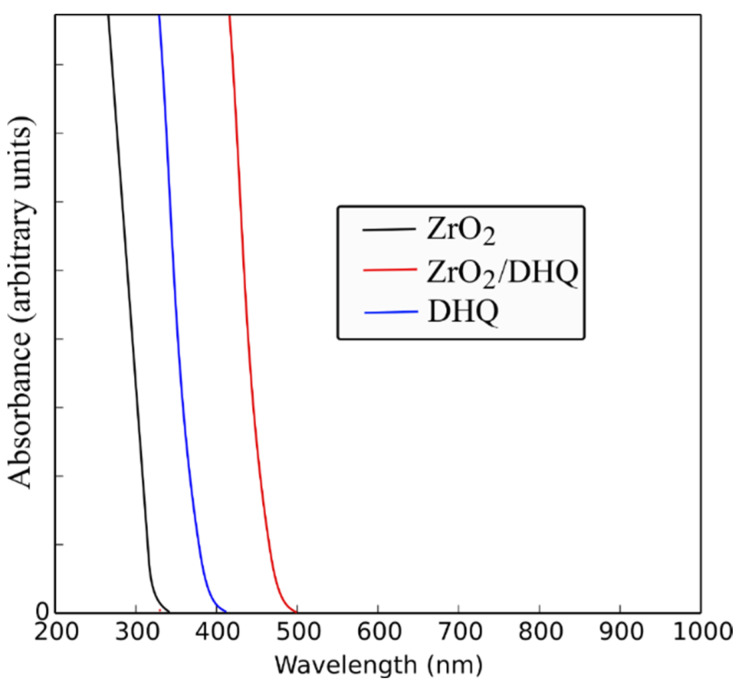
The electronic excitation spectra of [Zr_11_O_20_(OH)_2_]/DHQ cluster, and its constituents, calculated by convolution with a full width at a half maximum of 3000 cm^−1^.

**Figure 7 nanomaterials-12-03195-f007:**
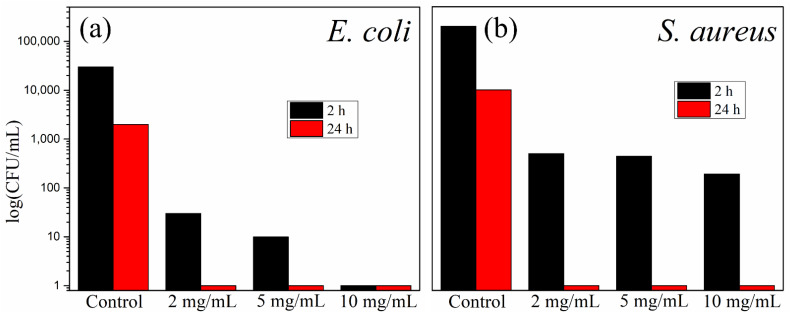
Concentration-dependent antimicrobial activity of ZrO_2_/DHQ/Ag against (**a**) *E. coli* and (**b**) *S. aureus*.

**Figure 8 nanomaterials-12-03195-f008:**
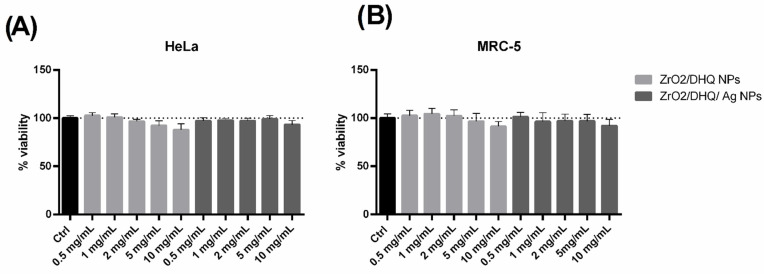
Viability of (**A**) HeLa and (**B**) MRC-5 cell lines as a function of the concentration of ZrO_2_/DHQ and ZrO_2_/DHQ/Ag.

**Table 1 nanomaterials-12-03195-t001:** The energies of frontier orbitals (eV) of DHQ, [Zr_11_O_20_(OH)_4_], and DHQ/[Zr_11_O_20_(OH)_2_] cluster, the bandgap (E_g_), as well as excitation energies, and the oscillator strength (ƒ) of the first three transitions calculated at TD-DFT/CAM-B3LYP/6-31G(d,p) level. Wave-functions and contributions to the electronic transitions are presented in the last column. HOMO/LUMO denote DHQ’s orbitals, while VBM and CBM are used for ZrO_2_-centered orbitals.

System	VBM	HOMO	CBM	LUMO	E_g_	Excitation Energy/nm (eV)	ƒ	Wave-Function (|Coefficient|^2^ ≥ 10%)
DHQ	-	−5.52	-	−1.70	3.82	363 (3.41)	0.51	HOMO → LUMO
[Zr_11_O_20_(OH)_4_]	−6.42	-	−2.65	-	3.77	308 (4.02)	0.02	VBM → CBM
						297 (4.17)	0.03	VBM-1 → CBM
						289 (4.28)	0.10	VBM → CBM+1
DHQ/[Zr_11_O_20_(OH)_2_]	−6.60	−5.84	−2.85	0.07	3.75	414 (2.99)	0.59	HOMO → CBM
						403 (3.08)	0.03	HOMO → CBM VBM → CBM
						383 (3.24)	0.02	VBM-2 → CBM

## Data Availability

Not applicable.

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
