# Peer review of "Toxicity of Silver Nanoparticles Supported by Surface-Modified Zirconium Dioxide with Dihydroquercetin"

_nanomaterials, 2022, doi:10.3390/nano12183195_

Round 1
Reviewer 1 Report
1- The title is informative and relevant; it could be more specific.
2- The introduction can be improved by providing a more critical discussion of recent related literature. Discuss the shortcomings of previous work and the gaps and how this work intends to fill those gaps. For example, some papers (Polymer Testing.1;93:106922), (Polymers. 9;12(4):861)
3- In the current state, there are some typographical errors. Therefore, the authors are advised to recheck the whole manuscript for improving the language and structure carefully.
4- The authors should prepare Figure 8 with better resolution.
5- References: The manuscript lacks the literature citation of some highly interesting most recent relevant works
Author Response
- The title is informative and relevant; it could be more specific.
Answer: We prefer to keep the title of the manuscript. In our opinion, the title is sufficiently specific since it reveals the nature of support, inorganic-organic hybrid, that plays a crucial role in the in-situ preparation of silver nanoparticles.
- The introduction can be improved by providing a more critical discussion of recent related literature. Discuss the shortcomings of previous work and the gaps and how this work intends to fill those gaps. For example, some papers (Polymer Testing.1;93:106922), (Polymers. 9;12(4):861).
Answer: As suggested by Reviewer 1, we extended the introduction to emphasize the significance of our research related to the current stage in this field (page 2, lines 47-53, ref 12). Both references, suggested by Reviewer 1, deal with Zn-doped MgO nanoparticles incorporated in hydrogels. Although these nanostructures are potentially applicable for biomedical purposes, we find these papers unrelated to our study because prepared composites did not contain silver, and an evaluation of the toxic action of synthesized samples wasn't carried out.
- In the current state, there are some typographical errors. Therefore, the authors are advised to recheck the whole manuscript for improving the language and structure carefully.
Answer: The paper reviewed the native English speaker, and, in addition, we carefully examined the manuscript to avoid typographical errors, and some small mistakes are rewritten.
- The authors should prepare Figure 8 with better resolution.
Answer: As suggested by Reviewer 1, we prepared Figure 8 in better resolution.
- References: The manuscript lacks the literature citation of some highly interesting most recent relevant works.
Answer: Most of the cited papers, out of 68, were published in the last 3-4 years. We made an additional effort to cover the most recent relevant work in this field and included in the revised version of the manuscript in the reference list a recent review paper, reference 12. Considering the lack of information in the literature concerning the toxicity of silver, prepared on the wide-bandgap metal-oxides-based hybrids displaying ICT-complex formation, we think that recent relevant work in this field is well-covered.
Reviewer 2 Report
The manuscript is well written and it addresses all necessary aspects related to the materials synthesis, characterization and properties. The subject is of high interest for the scientific community. No flaws were identified.
Author Response
The manuscript is well written and it addresses all necessary aspects related to the materials synthesis, characterization and properties. The subject is of high interest for the scientific community. No flaws were identified.
Answer: We thank Reviewer 2 for finding the manuscript suitable for publication as is.
Reviewer 3 Report
In this manuscript written by D. Sredojević et al. the authors have studied the antibacterial activity and cytotoxicity evaluation using Human Cervical Adenocarcinoma and human fetal lung fibroblast cell lines of silver nanoparticles on inorganic-organic hybrid based on zirconium dioxide nanoparticles and dihydroquercetin. This study is interesting and attractive for the readers in the fields of biology and medicine. Overall, it is an interesting, well organized and concise manuscript.
My comments are below:
1. In the introduction section, the authors should briefly present some of the existing relevant work emphasizing the novelty of the current study.
2. Fig. 7-What is the error involved in the evaluation of antimicrobial activity?
Author Response
Comments and Suggestions for Authors
In this manuscript written by D. Sredojević et al. the authors have studied the antibacterial activity and cytotoxicity evaluation using Human Cervical Adenocarcinoma and human fetal lung fibroblast cell lines of silver nanoparticles on inorganic-organic hybrid based on zirconium dioxide nanoparticles and dihydroquercetin. This study is interesting and attractive for the readers in the fields of biology and medicine. Overall, it is an interesting, well organized and concise manuscript.
Answer: We thank Reviewer 3 for finding the manuscript interesting for a broad audience in biology and medicine.
My comments are below:
- In the introduction section, the authors should briefly present some of the existing relevant work emphasizing the novelty of the current study.
Answer: As suggested by Reviewer 3, we extended the introduction (page 2, lines 47-53, ref 12) to better emphasize the novelty of our study.
- Fig. 7-What is the error involved in the evaluation of antimicrobial activity?
Answer: We thank Reviewer 3 for this comment. So, in the experimental, we included the additional section (2.5) concerning the statistical analysis of data (page 5, lines 210-215). The data were analyzed using the ANOVA one-way analysis of variance with the Tukey posthoc test (α = 0.05). The results are the mean values of three measurements. The scale is logarithmic, so the standard deviation was not visible.